# Length of stay to recover from severe acute malnutrition and associated factors among under-five years children admitted to public hospitals in Aksum, Ethiopia

**Wagnew Tesfay[1], Mebrahtu Abay[2], Solomon Hintsa[2], Tekia Zafu[2]***

**1** Medecins Sans Frontieres Holland, Ethiopia Mission, Tigray Project, **2** Department of Epidemiology and Biostatistics, School of Public Health, College of Health Science, Aksum University, Aksum, Ethiopia

* tekish2014@gmail.com

**Data Availability Statement:** All relevant anonymized data present within the manuscript as supplementary file S1.

## Abstract

### Background

Severe acute malnutrition is defined by <70% weight for length/height, by visible severe wasting, by the presence of pitting edema, and in children 6 to 59 months of age, mid upper arm circumference <110 mm. Severe acute malnutrition remains to be a worldwide problem, claiming lives of millions of children, especially in sub-Saharan Africa and south Asia. Though the Ethiopian national guideline states the total length of stay in therapeutic feeding units should not be more than four weeks, there is huge difference, varying from 8 to 47 days of stay. Therefore, the objective of this study was to assess length of stay to recover from severe acute malnutrition and associated factors among under five children hospitalized to the public hospitals in Aksum Town.

### Methods

Sample size was calculated using STATA version 12.0. A retrospective cohort study was conducted using pretested questionnaire in the public hospitals in Aksum on children aged 0–59 months. Cleaned data was entered to Epi info version 7.1.4 and then exported into SPSS version 21 for analysis. Bivariable and multivariable analyses were performed using Kaplan Meier and Cox regression models. During bivariable analysis, variables with p-value < 0.05 were selected for multivariable analysis to identify independent factors associated with length of stay.

### Results

A total of 564 participants enrolled to the study. The rate of recovery was 56% with median length of stay of 15 days (95% CI: 14.1, 15.9). The independent predictors of length of stay to recovery were presence of diarrhea at admission (AHR = 0.573, 95% CI: 0.415–0.793), being HIV positive (AHR = 0.391, 95% CI: 0.194–0.788), palmar pallor (AHR = 0.575, 95% CI: 0.416–0.794), presence of other co-morbidities at admission (AHR = 0.415, 95% CI: 0.302–0.570) and not being treated with plumpy nut (AHR = 0.368, 95% CI: 0.262–0.518).

**Funding:** The authors received no specific funding for this work.

**Competing interests:** The authors have declared that no competing interests exist.

## Conclusions

Length of stay is in the acceptable range of the international and national set of standards. Nevertheless, the recovery rate was lower compared to the Sphere standard. Presence of diarrhea, palmar pallor, HIV other co-morbidities and not treated with plumpy nut were found independent protective factors for recovery from sever acute malnutrition.

## Introduction

Severe acute malnutrition (SAM) is defined by <70% weight for length/height (WFL/H), by visible severe wasting, by the presence of pitting edema, and in children 6 to 59 months of age, mid upper arm circumference (MUAC) <110 mm [1–3]. Although there are basic and under-lying causes, SAM is an immediate effect of inadequate dietary intake (quality or quantity) and/or infections like tuberculosis (TB), human immune deficiency virus/acquired immune deficiency syndrome (HIV/AIDS) and diarrhea that often lead to nutrient mal-absorptions [4].

Although SAM usually affects all segments of a population, infants and young children are most vulnerable as they have higher nutritional requirements for growth and development [5]. It is one of the leading causes of morbidity and mortality among infants and young children all over the world and more frequently in sub-Saharan Africa and south Asia [6]. The peak age for SAM is 6–18 months, which is the time of fast growth and brain development. However, it is increasingly becoming common that SAM may occur in infants less than six months of age with many disadvantaged populations while starting to feed semi-solid and solid foods to children as young as two months [3].

The 2013 Ethiopian guideline for SAM management states that the total length of stay (LOS) in therapeutic feeding units (TFUs) should not be more than four weeks [7]. But there is huge difference in LOS in studies done in Ethiopia, varying from 8 to 47 days [8–10].

Severe acute malnutrition is a worldwide problem and one of the top deadly diseases for children less than five years of age. Severely malnourished children have a nine times more mortality rates than well-nourished children [11]. As per the 2018 United Nations' International Children's Emergency Fund, World Health Organization and World Bank report, wasting threatened lives of an estimated 50.5 million children under five globally, of which 13.8 million are from Africa[12], and also there are 52 million (about 8.3%) children less than five year suffering from acute malnutrition; out of those affected more than 90 percent are from South and Southeast Asia, and Sub-Saharan Africa [13].

Globally, more than ten million children die due to SAM before celebrating their fifth birthday every year. Huge number of severely malnourished children die at their home without any hospital care, but sometimes even hospital care is given, death rates may be high [7]. In developing countries, under five years children, who are severely malnourished and admitted to hospital, with unacceptably high case fatality rate (30–50%)[14].

In the developing world, where SAM is the most common reason for pediatric hospitalization, it is associated with higher risk of morbidity and mortality, underlying for more than 50 percent of the ten to eleven million children under five years old who die every year from avoidable causes [2, 15, 16]. Despite of such worldwide significance, child recovery programs have not given the required attention for facility based management of SAM [2].

According to studies from Kenya and Niger, recovery rate and LOS are affected by co-morbidities like pneumonia, malaria, altered consciousness, weak pulse, inability to drink,

temperature gradient, chest in-drawing, diarrhea & severe pallor [17, 18]. The horn of Africa has identified 6.6 per cent of children less than five years old as wasted. With estimated 10%, Ethiopia has highest rate of wasting (acute malnutrition) in this region [19]. Even though it was not well known in this particular study area, some studies in Ethiopia indicated that there is a big variation in LOS in Ethiopia, ranging from 8 to 47 days [8–10], that is not congruent with the National SAM management protocol that states the total LOS in TFUs should not be more than four weeks [7]. A study in northwest Ethiopia also revealed that patients admitted with SAM had a case fatality and defaulter rates of 18%, and 9% respectively [20].

UNICEF has conducted an evaluation study in Ethiopia [21] and identified gaps and management errors in the services like not giving routine medicines such as Amoxicillin, children who should be managed in phase two were managed in phase one, and those who could be treated in Outpatient Therapeutic Program (a program where moderately malnourished children get treatment) were being treated in Therapeutic Feeding Unit (TFU), poor medical record handling, late transfer of cases from phase to phase & to discharge, and limited area allowed to the TFU that increase cross infection and raised risk of death, especially for HIV infected children as.

Though the Federal Ministry of Health (FMoH), jointly with its partners, is taking promising actions in reducing SAM, the Ethiopian national underweight and wasting rates among under five children are still high, 24 and 10% respectively. The same is true in Tigray region as the underweight and wasting rates are 23 and 11% respectively [22, 23].

There were couples of researches conducted on malnutrition in the study area, but none of them attempted to assess the length of stay to recover from SAM and associated factors among under five children hospitalized to public hospitals in Aksum town, northern Ethiopia. So, this research was intended to come up with information about length of stay till recovery and associated factors among severely malnourished children admitted to the identified health facilities for the betterment of quality of care. The purpose of this study was therefore, to assess the length of stay to recover from SAM and identify associated factors among under five children admitted to the public hospitals in Aksum, north Ethiopia.

Moreover, coming up with recent information about length of stay till recovery and associated factors among severely malnourished children is enormously relevant to be used as input by clinicians for betterment of quality of care given to clients and as a guide (reference) for academicians who are interested to conduct further related research on the area.

## Methods and materials

### Study setting, design and ethical issue

Ethical clearance and permission letter were taken from the Institutional Review Committee (IRC) of Aksum University, College of health science. For the secondary data from patients' medical registration cards, the IRC waived the requirement for informed consent.

Health facility based retrospective cohort study was conducted in the two public hospitals of Aksum Town, Ethiopia. Aksum is found in Tigray regional state, 1042 km far from the capital city of Ethiopia, Addis Ababa. St. Mary general hospital and Aksum University comprehensive specialized hospital are the two public hospitals found in Aksum town, providing inpatient and outpatient services for approximately 2.8 million catchment population. Medical and health officer (HO) interns, nurses, general practitioners (GPs) and a pediatrician in each hospital are providing medical care and treatment for SAM children admitted to both health institutions. Admission, management and discharge procedures for SAM are compatible with the Protocol developed by Federal Ministry of Health (FMoH), that is updated in 2013 [7].

The study was carried out on those admitted with SAM from September 11, 2016 to February 10, 2019. Data was extracted within 15 days (from March 6–20, 2019).

## Source and study population

The source populations were all severely malnourished children, aged 0–59 months, admitted to TFUs of the public hospitals in Aksum and the study populations were all severely malnourished children, aged 0–59 months, admitted to TFUs of the public hospitals in Aksum from September 11, 2016 to February 10, 2019.

## Inclusion and exclusion criteria

Based on the Ethiopian Federal Mistry of Health National protocol for SAM management [7], 1) Infants < 6 months of age or <3 kilogram with median Weight-For-Length of <70% and/ or presence of bilateral pitting edema, or visible severe wasting. 2) Children aged 6–59 months with median Weight-For-Length/Height of <70%, and/or bilateral pitting edema, or MUAC <11Centimeter were included to the study. Children whose admission and discharge date was not recorded, as this will not show the outcome variable (LOS), with congenital anomalies (like cleft lip and developmental disorders such as Down's syndrome), as these can be a confounding with LOS for SAM, and those whose medical records were incomplete for the type of SAM were excluded from the study.

**Sample size and sampling procedure.** Sample size was calculated based on sample size estimation for survival under the Cox proportional hazards model using the STATA Version 12.0, taking the following assumptions into account: a 1.8 adjusted hazard ratio, 17.1% (0.171) observed prevalence of failure/non-recovery [24], with marginal error of 5% and confidence level of 95%. With the assumption of number of study subjects required to achieve a study power of 80% and 10% incompleteness, the overall sample size became 585.

The actual number of under five children admitted with SAM to the stated hospitals in the study period was estimated to be 700 (280 from Aksum University comprehensive specialized hospital and 420 from St. Mary hospital). Systematic random sampling technique was used to select the 585 study participants from the actual 700 taking each participant consecutively (700/585 = 1.2). With this proportion, 234 participants were selected consecutively from Aksum University comprehensive specialized hospital (AKU-CSH) and 351 from St. Mary General hospital. Out of these eligible study participants, 564 of them were enrolled into the study whereas the remaining 21 (3.6%) children were excluded. Among these excluded, the clinical files for 12 children from St. Mary hospital were not found and one participant was removed during analysis as it was influential extreme value from same institution whereas total of eight children from AKU-CSH were excluded (four were admitted with congenital anomalies and developmental disorders (one with cleft lip and three with Down's syndrome) and clinical files were incomplete for type of SAM for the remaining four).

## Variables and measurement

**Dependent/outcome variable.** Time till recovery (LOS): refers to the number of days/ weeks it takes from hospitalization till when a child recovered from SAM of any kind. Children are called recovered when they got relieved from medical complications, edema and have gained and maintained WFL/WFH of 85% [7].

Event: recovery from sever acute malnutrition

Censored: Not recovered from SAM. (Death, defaulter, medical referral, transferred)

Death: when the SAM child die while receiving treatment in the TFU and registered as dead in the treatment logbook [7].

Defaulters: those who were not found in the NRU for two successive days, or who leave the ward against professional advice while the child is not cured [7, 13, 25].

**Independent variables.**

- Severe acute malnutrition diagnosis (marasmus, kwashiorkor marasmic-kwashiorkor)

- Study subject characteristics (age, sex, residence)

- Routine medications, supplements and therapeutic feedings

- Common comorbidities (HIV, Anemia, Diarrhea)

- Other co-morbidities (like malaria, pneumonia, tuberculosis)

## Data collection tools and data quality control

A compilation sheet (coding check list) was developed relating to the FMoH standard management protocol for SAM. Then needed individual data was extracted from relevant documents like SAM registration logbook, SAM monitoring multi-chart and patient clinical files.

The compilation sheet (check list) was prepared in English was and then pre-tested in Sehul general hospital, Shire town, Ethiopia in 29 participants (5% of the total sample) and revised for sequence and layout. Four Bachelor of Science degree holder health care professionals were recruited as data collectors with one supervisor. They got one day training on how to fill the check list to minimize errors. The collected data were checked by the principal investigator closely for its accuracy, completeness and consistency.

## Data management and analysis process

Then collected data were cleaned, coded and entered to Epi info version 7.1.4.0 and then exported to SPSS version 21 for analysis. The levels of missing values, existence of influential extreme values and multi co-linearity among independent variables was checked before analysis. Descriptive analysis was performed using frequency and median, and presented using tables. Kaplan-Meier & Cox regression models were applied to determine the association of independent variable with length of stay to recovery from SAM. P-value of $< 0.05$ was taken as statistically significant to identify explanatory factors of LOS in multivariable analysis.

Thirty one independent variables were analyzed in the bivariable Cox proportional hazard regression analysis; out of which twenty one were suggestive of significance (p-value $< 0.05$) and entered to multivariable analysis. A stepwise forward likelihood ratio variable selection method, with entry and removal probabilities of 0.05 and 0.1, respectively, was made on significant factors in the bivariable analysis. Accordingly, the final Cox Proportional Hazards Model was fitted on the basis of final step selected variables after model assumption diagnostic procedures. Remedial measures were taken on potential problems found in checking validity of assumptions. In due course, neither multi-collinearity problem nor significant interaction was detected. However, examination of *Dfbeta* statistics in relation to model coefficients indicated the existence of influential extreme value. Thus, the outlier was found to be the case entry with unique SAM serial number 411 and its removal resulted in magnitudes of 0.16 and 0.26 change to the corresponding model coefficients of HIV Status. As a result, the final model was fitted on 564 observations of selected variables.

The proportionality of hazards assumption was further checked by examining plots of recovery time for model variables. Patterns of the plot were not seemed to cross each other, rather were close to be parallel. This left no doubt about violation of proportionality of hazards

assumption. Further this was ascertained by examination of cumulative hazard plot of the Cox-Snell residual. The plotted points nearly lie around a line that has unit slope and zero intercept, which also confirmed validity of proportional hazards assumption. Finally, goodness of the overall fit of the model was assured by Omnibus tests of model coefficients at 5% level of significance.

## Results

Out of total 585 eligible severely malnourished children hospitalized to the public hospitals of Aksum, data from 564 children were extracted with retrieval rate of 96.4%. Out of them 56% were recovered from SAM, and the recovery times for the remaining 44% were considered to be censored at the time of analysis.

### Socio-demographic characteristics, co-morbidity and type of SAM

More than half (53.9%) of the children enrolled into the study were males with the corresponding median recovery time of 15 days (95% CI: 13.7, 16.3), similar with the overall median LOS of 15 days (95% CI: 14.1, 15.9).

Age was grouped into categories in convenience for analysis. Median survival time was measured in days by sex and age categories of the study participants. The largest percentage of children was categorized under the age group of 12–35 months which accounted for 54.6% of the total subjects with an estimated median recovery time of 15 days in a 95% CI (13.7, 16.3). In contrast, the percentage of children under age group 0–5 months were only 10.8%, the least in percentage comparison despite its association with the highest median recovery time of 20 days with a 95% CI (16.4, 23.6).

The median recovery time for children with Marasmus which accounted for 64.9% of the study participants was found to be 15 days with 95% CI (13.8, 16.2), the same point estimate with those having Kwashiorkor (20.7% of the total). The remaining 14.4% were children with Marasmic-Kwashiorkor having, 17 days of average length of recovery time (95% CI; 13.6, 20.4) (Table 1).

### Treatment, treatment outcomes, routine medications and feedings

Out of the participants enrolled, 56% were discharged cured, 22.2% transferred to nearby health facility, 9.8% defaulted, 6% dead during treatment and 6% were medical referral. 18.3 And 8 per cent of participants received IV fluid and blood transfusion respectively. In Bivariate analysis, not receiving both IV fluid and blood transfusion found significantly associated with shorter LOS till recovery and both were selected for multivariate analysis to check if predictors of LOS to recover from SAM.

Almost all (98.8%) received antibiotics like amoxicillin, ampicillin and gentamycin and higher numbers of children were supplemented folic acid and iron (68.1% and 61.9% respectively). Regarding feedings, F75 was offered for almost all patients (99.8%) followed by F100 (for 81% of participants) and half of them were given plumpy nut. Meanwhile, 31.9 per cent were fed with NG tube. As receiving folic acid, vitamin A, F100, plumpy nut and not being fed with NG tube were suggestive of shorter recovery time in bivariate analysis, these all were selected for further multivariable analysis using Cox regression model. For those discharged cured, the average weight gain was 10.1g/kg/day (Table 1). The overall median recovery time from SAM was also determined to be 15 days with a 95% confidence interval between 14.1 and 15.9 days (Fig 1).

**Predictors of length of stay to recovery from sever acute malnutrition.** After ascertaining validity of the model assumptions and adjustment, five independent significant predictors

**Table 1.  Log rank (Mantel-Cox) test of equality of survival distribution for socio-demographic characteristics, type of SAM and co-morbidity at admission in severely malnourished children admitted to public hospitals in Aksum, 2019.**

| Variable | Category | Percentage of outcome category | | | | | | Median Survival (days) | 95% CI | | Log-rank test | |
|---|---|---|---|---|---|---|---|---|---|---|---|---|
| | | Cured % | Dead % | Defaulted % | Transferred % | Referred % | Total % | | LL | UL | Chi-square | P-value |
| Sex | Male | 31.4 | 2.3 | 4.8 | 12.2 | 3.2 | 53.9 | 15 | 13.7 | 16.3 | 2.87 | 0.090 |
| | Female | 24.6 | 3.7 | 5.0 | 9.9 | 2.8 | 46.1 | 16 | 14.6 | 17.4 | | |
| Age (in months) | 0–5 | 7.3 | 1.1 | 0.7 | 1.1 | 0.7 | 10.8 | 20 | 16.4 | 23.6 | 8.74 | 0.033 |
| | 6–11 | 10.1 | 0.5 | 2.1 | 5.5 | 1.6 | 19.9 | 14 | 13.2 | 14.8 | | |
| | 12–35 | 29.8 | 3.4 | 5.5 | 12.9 | 3.0 | 54.6 | 15 | 13.7 | 16.3 | | |
| | 36–59 | 8.9 | 1.1 | 1.4 | 2.7 | 0.7 | 14.7 | 15 | 13.1 | 16.9 | | |
| Name of institution | AkUCSH | 23.2 | 2.3 | 4.6 | 6.0 | 3.9 | 40.1 | 16 | 14.1 | 17.9 | 7.34 | 0.007 |
| | St. Mary hospital | 32.8 | 3.7 | 5.1 | 16.1 | 2.1 | 59.9 | 15 | 13.9 | 16.1 | | |
| Type of SAM | Marasmus | 35.5 | 3.5 | 6.9 | 14.9 | 4.1 | 64.9 | 15 | 13.8 | 16.2 | 0.384 | 0.825 |
| | Kwashiorkor | 11.2 | 1.1 | 2.0 | 5.1 | 1.4 | 20.7 | 15 | 13.1 | 16.9 | | |
| | Marasmic-kwash | 9.4 | 1.4 | 0.9 | 2.1 | 0.5 | 14.4 | 17 | 13.6 | 20.4 | | |
| Co-morbidity | Present | 38.8 | 6.0 | 5.0 | 8.7 | 5.9 | 64.4 | 17 | 15.8 | 18.2 | 82.82 | <0.001 |
| | Absent | 17.2 | 0.0 | 4.8 | 13.5 | 0.2 | 35.6 | 11 | 10.0 | 12.0 | | |
| Total | | 56.0 | 6.0 | 9.8 | 22.2 | 6.0 | 100.0 | 15 | 14.1 | 15.9 | | |

of LOS for nutritional recovery were found, which are; diarrhea, HIV sero-status, palmar pallor (anemia), other co-morbidities (pneumonia, TB and Malaria), and provision of plumpy nut. Patients with diarrhea at admission were 42.7% (AHR = 0.573; 95% CI: 0.415–0.793) less likely to recover quickly from SAM as compared to those without diarrhea while those HIV positive found to be 60.9% (AHR = 0.391; 95% CI: 0.194–0.788) less likely to get cured fast in comparison with those whose sero-status is unknown. However, no significant difference was obtained in the hazards of non-recovery for children with Non-reactive (AHR = 0.937; 95%

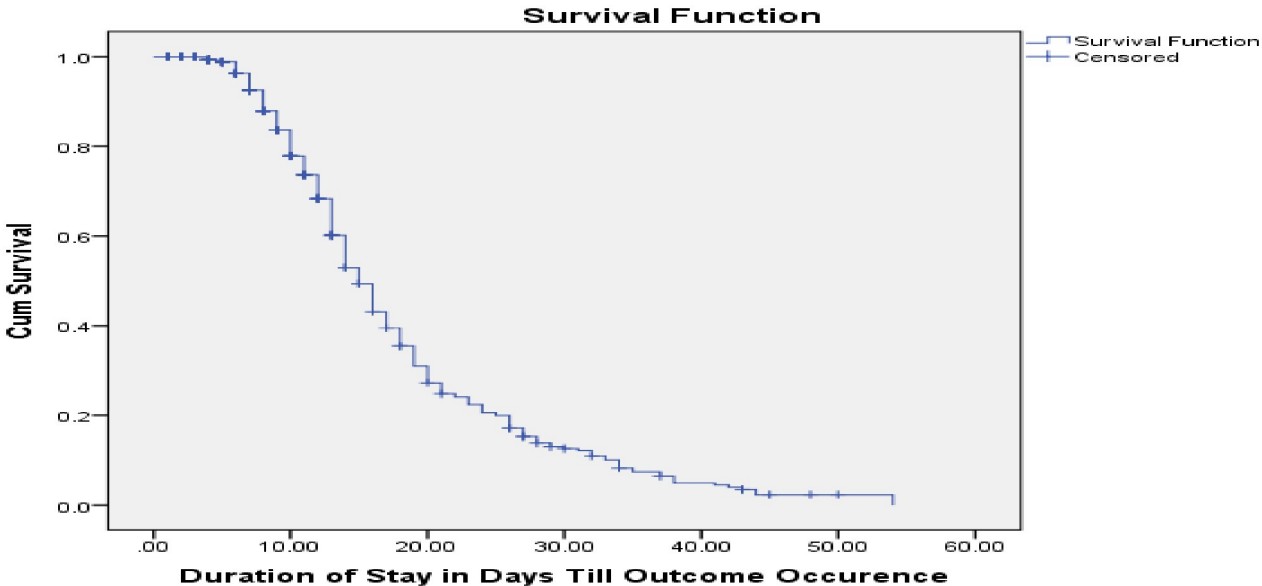

**Fig 1. Overall survival function of children with SAM admitted to public hospitals in Aksum, 2019.**

CI: 0.714–1.231) and unknown HIV status. The difference in chance of recovery between reactive and non-reactive patients stays significant at 5% level of significance. Children with palmar pallor (anemia) and other co-morbidity at admission were 42.5% (AHR = 0.575; 95% CI: 0.416–0.794) and 58.5% (AHR = 0.415; 95% CI: 0.302–0.570) less likely to recover earlier as compared to those who did not have such conditions. Patients who were not treated with plumpy nut were 63.2% (AHR = 0.368; 95% CI: 0.262–0.518) less likely to recover in comparison with those who received plumpy nut (Table 2).

## Discussion

Results of this study were examined for consistency with extensive review of literatures of the title. At admission, 64.4% of participants showed up with other co-morbidities on top of SAM. Majority of children (64.9%) were hospitalized with marasmus (non-edematous) type of SAM, similar with some recent studies in Ethiopia [1, 13, 25–31] but in contrary to that of Jimma [32], Hadiya, Ethiopia [33] and Uganda [34], which reported edematous type of SAM was highly encountered. This variation might be attributable to the multi-faceted causes of SAM all over the world.

The median LOS till recovery from SAM was estimated to be 15 days (95% CI: 14.1, 15.9), which is in the acceptable range of international standards set by the SPHERE project [15]. This is similar with the findings from institution based researches in Bahirdar [13, 35] that reported 16 days of recovery time. But this finding is far less than the report from Sidama zone Shebedino district of southern Ethiopia [29] which found the median LOS of 36 days. It is also lower than several studies conducted in the Ethiopia [1, 8, 11, 24, 32] and that of Yemen [36]. However, the median LOS is higher than some study reports from some parts of Ethiopia [25, 26, 33], Ghana [20] and India [37]. This could be due to the differences in underlying co-morbidities, caring practice of healthcare providers, health facility set up and variation in socioeconomic status of the population in these different study areas.

The overall rate of recovery from SAM was found to be 56%, that is consistent with findings from Debrebrhan University that revealed 55.9% rate of recovery [37]. It is significantly higher than that of Ayder hospital (11) other similar studies from Bahirdar [13, 35], Ghana [20] and Yemen [36]. Nevertheless, it is by greater margin below the minimum international standards [15], in comparison with other study findings in Ethiopia [1, 8, 24, 25, 27–32] and similar reports from India, Malawi and Uganda [34, 37–39] as well. This might be due to the relatively higher rate of transfer out to nearby health facility, which probably would be to prevent patient overload since one of the study area (AKU-CSH) is the only referral hospital in this particular study area. Almost a quarter (22.2%) of patients were transferred out to nearby health facility that is above other recently reported study findings in Ethiopia [1, 13, 25, 27, 35] Ghana [20], Uganda [34] and Yemen [36].

This study shows death rate of 6% from the total participants enrolled, which is acceptable by the SPHERE project minimum international standards for managing SAM in NRU/TFU, and better than the recent study findings from University of Gondar [28] and Hawassa University [31] comprehensive specialized hospitals which found mortality rates of 10.8% for each. It is similar with the findings of two studies done in Ethiopia [25, 26] and one conducted in Nigeria [40]. But the mortality rate is greater than that of some studies done in Ethiopia [11, 13, 35], India [37, 38], Ghana [20] and Yemen [36]. This could be due to lack of close follow up of patients with strict adherence to the national or international SAM management protocols and socioeconomic differences in the different areas.

According to the multivariate analysis, there was significant difference in median LOS till cured from SAM among predictor variables. Participants who showed up with palmar pallor

**Table 2. Factors associated with LOS to recover from SAM in children admitted to public hospitals in Aksum, 2019.**

| Variable | Category | Recovery status | | | | CHR (95% CI) | P-value | AHR (95% CI) |
|---|---|---|---|---|---|---|---|---|
| | | Non-recovered | | Recovered | | | | |
| | | N | % | N | % | | | |
| Age | 0–5 | 20 | 3.5 | 41 | 7.3 | 0.639(0.421, 0.969) | 0.048* | 0.702(0.369, 1.334) |
| | 6–11 | 55 | 9.8 | 57 | 10.1 | 1.105(0.755, 1.616) | | 1.102(0.711, 1.706) |
| | 12–35 | 140 | 24.8 | 168 | 29.8 | 0.982(0.715, 1.348) | | 1.012(0.717, 1.428) |
| | 36–59 | 33 | 5.9 | 50 | 8.9 | 1 | | 1 |
| Name of institution | AKU-CSH | 95 | 16.8 | 131 | 23.2 | 1.354(1.077, 1.702) | 0.01* | 0.775(0.573, 1.048) |
| | St. Mary hospital | 153 | 27.1 | 185 | 32.8 | 1 | | 1 |
| Other Co-morbidities | Present | 144 | 25.5 | 219 | 38.8 | 3.103(2.384, 4.039) | <**0.001**** | 0.415(0.302, 0.570) |
| | Absent | 104 | 18.4 | 97 | 17.2 | 1 | | 1 |
| Diarrhea | Present | 166 | 29.4 | 235 | 41.7 | 1.973(1.518, 2.565) | <**0.001**** | 0.573(0.415, 0.793) |
| | Absent | 82 | 14.5 | 81 | 14.4 | 1 | | 1 |
| Vomiting | Present | 141 | 25.0 | 189 | 33.5 | 1.509(1.198, 1.901) | <0.001* | 0.883(0.684, 1.138) |
| | Absent | 107 | 19.0 | 127 | 22.5 | 1 | | 1 |
| Fever | Present | 33 | 5.9 | 32 | 5.7 | 1.515(1.049, 2.187) | 0.027* | 1.066(0.695, 1.637) |
| | Absent | 215 | 38.1 | 284 | 50.4 | 1 | | 1 |
| Hypothermia | Present | 39 | 6.9 | 43 | 7.6 | 1.929(1.391, 2.674) | <0.001* | 0.767(0.529, 1.112) |
| | Absent | 209 | 37.1 | 273 | 48.4 | 1 | | 1 |
| HIV status | Reactive | 5 | 0.9 | 13 | 2.3 | 0.314(0.091, 0.583) | **0.004**** | 0.391(0.194, 0.788) |
| | Non-reactive | 107 | 19.0 | 174 | 30.9 | 0.984(0.087, 0.232) | | 0.937(0.714, 1.231) |
| | Unknown | 136 | 24.1 | 129 | 22.9 | 1 | | 1 |
| Pulse rate | Bradycardic | 4 | 0.8 | 3 | 0.6 | 0.559(0.445, 0.702) | <0.001* | 2.227(0.676, 7.336) |
| | Normal | 124 | 23.5 | 160 | 30.3 | 0.363(0.054, 0.762) | | 1.363(1.054, 1.762) |
| | Tachycardic | 101 | 19.1 | 136 | 25.8 | 1 | | 1 |
| Consciousness level | Conscious | 142 | 25.2 | 186 | 33.0 | 0.492(0.393, 0.616) | <0.001* | 14.29(0.00, 93.70) |
| | Lethargic | 93 | 16.5 | 130 | 23.0 | 10.49 (6.78, 13.34) | | 11.53(0.00, 75.59) |
| | Comatose | 13 | 2.3 | 0 | 0.0 | 1 | | 1 |
| Palmar pallor | Present | 89 | 15.8 | 122 | 21.6 | 2.036(1.610, 2.574) | <**0.001**** | 0.575(0.416, 0.794) |
| | Absent | 159 | 28.2 | 194 | 34.4 | 1 | | 1 |
| Skin lesion | Present | 41 | 7.3 | 82 | 14.5 | 1.80(1.389, 2.332) | <0.001* | 0.982(0.697, 1.384) |
| | Absent | 207 | 36.7 | 234 | 41.5 | 1 | | 1 |
| Dehydration | Present | 148 | 26.2 | 231 | 41 | 2.143(1.645, 2.791) | <0.001* | 1.107(0.802, 1.528) |
| | Absent | 100 | 17.7 | 85 | 15.1 | 1 | | 1 |
| Shock | No | 233 | 41.3 | 310 | 55 | 0.196(0.080, 0.481) | <0.001* | 1.481(0.584, 3.755) |
| | Yes | 15 | 2.7 | 6 | 1.1 | 1 | | 1 |
| Folic acid | No | 111 | 19.7 | 69 | 12.2 | 1.427(1.090, 1.868) | 0.010* | 1.003(0.670, 1.502) |
| | Yes | 137 | 24.3 | 247 | 43.8 | 1 | | 1 |
| Vitamin A | No | 110 | 19.5 | 76 | 13.5 | 1.575(1.214, 2.043) | 0.001* | 0.881(0.635, 1.223) |
| | Yes | 138 | 24.5 | 240 | 42.6 | 1 | | 1 |
| F-100 | No | 83 | 14.7 | 24 | 4.3 | 1.639(1.077, 2.493) | 0.021* | 1.284(0.779, 2.117) |
| | Yes | 165 | 29.3 | 292 | 51.7 | 1 | | 1 |
| Plumpy nut | No | 226 | 40.1 | 56 | 9.9 | 2.774(2.073, 3.711) | <**0.001**** | 0.368(0.262, 0.518) |
| | Yes | 22 | 3.9 | 260 | 46.1 | 1 | | 1 |

Other comorbidities: Malaria, TB, Pneumonia * = significant at p-value < 0.05, Length of stay (LOS)

(indicative of anemia) at admission were by 42.5 per cent (AHR = 0.575; 95% CI: 0.416–0.794) less likely to have fast recovery than those who did not have pallor. This is in line with the findings reported from Nekemte and Bahirdar Felegehiwot referral hospitals, Ethiopia [10, 13]. Similarly, not receiving plumpy nut (AHR = 0.368; 95% CI: 0.262–0.518) was observed as strong independent predictor of recovery time, which also is consistent with report from Bahirdar [13]. But other related studies from Ethiopia [9, 32] reported that neither palmar pallor nor plumpy nut as independent predictor of LOS till recovery. This could be due to inter institutional differences in strictly adhering to the national SAM management guideline.

In line with this research findings, studies from different parts of the country [1, 10, 25] and studies from Malawi and Uganda [34, 39] reported that children presented with retroviral infection at hospitalization were less likely to recover from SAM, as being reactive for HIV serostatus among the study participants had negatively affected LOS to recover from SAM (AHR = 0.391; 95% CI: 0.194–0.788).

Probability of getting cured fast was reduced by 58.5% (AHR = 0.415; 95% CI: 0.302–0.570) in those admitted with co-morbidity. Consistent with this report, a hospital based retrospective cross-sectional study from Bahirdar and retrospective cohort study in similar setting in Jimma University found less recovery rate of SAM in co-morbid children [1, 32].

## Limitations of the study

Even though the strength of this paper comes from its study design (cohort), it totally was based on patients' secondary data, in which incompleteness was observed to some extent, and lacked control over the quality of measurements taken during hospitalization. It was also impossible to analyze socio-economic characteristics of parents/guardians and factors related to patient treatment (medical/pharmaceutical supplies and healthcare provider expertise) that could have influenced the outcome variable in a desirable or undesirable way.

## Conclusion

This research figured out that the median LOS till recovery is in the acceptable range of the national and international standards set to manage SAM as in patient. However, the rate of recovery was lower as compared to the stated standards and other study findings conducted in the nation. It also revealed significant differences in the median LOS to recover among different predictor variables. The study indicates that children that showed up with presence of diarrhea, palmar pallor, retroviral infection, other co-morbidities and those who did not receive plumpy nut have had lower chance of recovering from SAM.

## Supporting information

**S1 Data.**
(SAV)

## Acknowledgments

We are cordially thankful for the data collectors, the supervisor and staffs of St. Mary general hospital and Aksum University comprehensive specialized hospital for their support during data collection.

## Author Contributions

**Conceptualization:** Wagnew Tesfay.

**Data curation:** Wagnew Tesfay, Mebrahtu Abay, Solomon Hintsa, Tekia Zafu.

**Formal analysis:** Wagnew Tesfay, Mebrahtu Abay, Solomon Hintsa, Tekia Zafu.

**Funding acquisition:** Wagnew Tesfay.

**Investigation:** Wagnew Tesfay.

**Methodology:** Wagnew Tesfay, Mebrahtu Abay, Solomon Hintsa.

**Project administration:** Wagnew Tesfay.

**Resources:** Wagnew Tesfay, Mebrahtu Abay, Solomon Hintsa, Tekia Zafu.

**Software:** Wagnew Tesfay, Mebrahtu Abay, Solomon Hintsa, Tekia Zafu.

**Supervision:** Mebrahtu Abay, Solomon Hintsa, Tekia Zafu.

**Validation:** Wagnew Tesfay, Mebrahtu Abay, Solomon Hintsa.

**Visualization:** Wagnew Tesfay, Mebrahtu Abay, Solomon Hintsa, Tekia Zafu.

**Writing – original draft:** Wagnew Tesfay.

**Writing – review & editing:** Wagnew Tesfay, Mebrahtu Abay, Solomon Hintsa, Tekia Zafu.

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
