## [Decision Letter · Decision Letter 0]

23 Mar 2020

PONE-D-20-03753

Length of stay to recover from severe acute malnutrition and associated factors among under-five years children admitted to public Hospitals in Aksum, Ethiopia

PLOS ONE

Dear DR Tekia Zafu,

Thank you for submitting your manuscript to PLOS ONE. After careful consideration, we feel that it has merit but does not fully meet PLOS ONE’s publication criteria as it currently stands. Therefore, we invite you to submit a revised version of the manuscript that addresses the points raised during the review process.

We would appreciate receiving your revised manuscript by May 07 2020 11:59PM. To enhance the reproducibility of your results, we recommend that if applicable you deposit your laboratory protocols in protocols.io, where a protocol can be assigned its own identifier (DOI) such that it can be cited independently in the future. For instructions see: http://journals.plos.org/plosone/s/submission-guidelines#loc-laboratory-protocols

We look forward to receiving your revised manuscript.

Kind regards,

Shamala Devi Sekaran

Academic Editor

PLOS ONE

Journal Requirements:

2. In ethics statement in the manuscript and in the online submission form, please provide additional information about the patient records used in your retrospective study. Specifically, please ensure that you have discussed whether all data were fully anonymized before you accessed them and/or whether the IRB or ethics committee waived the requirement for informed consent. If patients provided informed written consent to have data from their medical records used in research, please include this information.

5. ** Please include your tables as part of your main manuscript and remove the individual files **. Please note that supplementary tables (should remain/ be uploaded) as separate "supporting information" files.

6. Please ensure that you refer to Figure 3 in your text as, if accepted, production will need this reference to link the reader to the figure.

Additional Editor Comments (if provided):

An extensive correction of the language is the first step

Seconding paper not formatted as for PLOS one and many areas incorrectly cited

PLease respond to all queries from both reviewers

Reviewers' comments:

Reviewer's Responses to Questions

**Comments to the Author**

1. Is the manuscript technically sound, and do the data support the conclusions?

Reviewer #1: Partly

Reviewer #2: Yes

2. Has the statistical analysis been performed appropriately and rigorously? 

Reviewer #1: I Don't Know

Reviewer #2: I Don't Know

3. Have the authors made all data underlying the findings in their manuscript fully available?

Reviewer #1: Yes

Reviewer #2: Yes

4. Is the manuscript presented in an intelligible fashion and written in standard English?

Reviewer #1: No

Reviewer #2: No

5. Review Comments to the Author

Reviewer #1: The manuscript has not been formatted or written according to the standard of PlosOne.

Some of the words/sentences used clearly reflects that this was taken directly from a thesis without careful revision by all the authors. This is ridiculous.

Although the study has revealed significant differences in the median LOS to recover among different predictor variables nevertheless, in general the findings are highlighting that the length of stay is in the acceptable range of the international and national set of standards. In addition the findings are more relevant to Hospitals in Askum and not sure if the findings will be impactful/relevant to other regions of Africa /countries experiencing SAM children. Probably this should be highlighted further

Reviewer #2: The aim as stated (line106) “this research was intended to come up with information about length of stay till recovery and associated factors among severely malnourished children admitted to the identified health facilities for the betterment of quality of care.” It was however, not clear whether the intention was to see “the effect of associated factors on the length of stay in the hospital” or “to find out what were the associated factors in children who were admitted with severe acute malnutrition”. Of course, on reading the research paper it becomes clear that the objective is to see the effect.

On page 17 under predictors of length of stay : “eight independent significant predictors of LOS for nutritional recovery were found, which are; diarrhea, HIV sero status, palmar pallor, co-morbidity, blood transfusion, IV fluid infusion, provision of plumpy nut and feeding with the help of NG tube”. I would think diarrhea, HIV sero status, palmar pallor should be included under “co-morbidity” and thus “co-morbidity” per se would not be an independent predictor of the length of stay.

Similarly, “deranged pulse rate”, “severe dehydration,” “altered consciousness”, “shock” and “hypothermia” under the heading “Clinical conditions of patients at admission(page 15, lines 295-308)” would all probably be parts of the same clinical spectrum at admission. These are all analyzed separately for incidence.

Since bilateral edema was seen in 90.5% of patients, it will be interesting to find out the cause of unilateral edema in the undernourished children studied (lines 303-304). It will be nice if this was mentioned too. This may not be related to the present research, but if it is due to cellulitis/ lymphadenitis etc. it would impact the length of stay for recovery.

While quick/ early recovery period from other complications (palmar erythema, co-morbidity, diarrhea) was calculated as percentage (of what? Of total, or those who did recover) the recovery in children who did not require NG tube feeding or I.V. fluids was calculated as “1.9 times and 1.55 times more likely”.

Early/quick recovery has not been defined (Line 337 & 344-45).

“Probability of getting cured fast was reduced by 58.5% (AHR = 0.415; 95% CI: 0.302-0.570) in those admitted with co-morbidity” (Line 424). Are the terms “recovery” and “cure” used interchangeably here?

Some abbreviations are not explained, e.g. “OTP (line96) and NRU (205) which I presume is the therapy unit where the children were admitted.

I thought the references should include the names of journals where they are originally published. They are of course easily traceable.

6. PLOS authors have the option to publish the peer review history of their article (what does this mean?). If published, this will include your full peer review and any attached files.

Reviewer #1: No

Reviewer #2: Yes: Professor Usha Rani Singh

---

## [Author Response · Author response to Decision Letter 0]

23 May 2020

See the attached response to reviewers letter

---

## [Editor Report · Decision Letter 1]

14 Aug 2020

Length of stay to recover from severe acute malnutrition and associated factors among under-five years children admitted to public Hospitals in Aksum, Ethiopia

PONE-D-20-03753R1

Dear Dr. Tekia Zafu,

We’re pleased to inform you that your manuscript has been judged scientifically suitable for publication and will be formally accepted for publication once it meets all outstanding technical requirements.

Kind regards,

Shamala Devi Sekaran

Academic Editor

PLOS ONE
---

## [Editor Report · Acceptance letter]

3 Sep 2020

PONE-D-20-03753R1

Length of stay to recover from severe acute malnutrition and associated factors among under-five years children admitted to public Hospitals in Aksum, Ethiopia

Dear Dr. Zafu:

I'm pleased to inform you that your manuscript has been deemed suitable for publication in PLOS ONE. Congratulations! Your manuscript is now with our production department.

Kind regards,

on behalf of

Professor Shamala Devi Sekaran 

Academic Editor

PLOS ONE